# Cervical Lymph Node Metastasis Differences in Patients with Unilateral or Bilateral Papillary Thyroid Microcarcinoma: A Multi-Center Analysis

**DOI:** 10.3390/jcm11164929

**Published:** 2022-08-22

**Authors:** Zheyu Yang, Yu Heng, Weihua Qiu, Lei Tao, Wei Cai

**Affiliations:** 1Department of General Surgery, Ruijin Hospital, Shanghai Jiaotong University School of Medicine, Shanghai 200025, China; 2ENT Institute and Department of Otorhinolaryngology, Eye & ENT Hospital, Fudan University, Shanghai 200031, China; 3Department of General Surgery, Civil Aviation Shanghai Hospital, Shanghai 200051, China

**Keywords:** papillary thyroid microcarcinoma, central lymph node metastasis, lateral lymph node metastasis, bilateral disease, unilateral disease

## Abstract

**Purpose****s:** To quantitatively predict the risk of neck lymph node metastasis for unilateral and bilateral papillary thyroid microcarcinomas (PTMC) that may guide individual treatment strategies for the neck region. **Methods:** A total of 717 PTMC patients from three medical centers were enrolled for analysis. **Results:** Bilateral PTMCs were demonstrated to be more aggressive with a much higher cervical lymph node metastasis rate including for both central (CLNM) and lateral lymph node metastasis (LLNM) when being compared to unilateral PTMCs. In unilateral PTMC, five (age < 55 years old, male, maximum tumor diameter (MTD) ≥ 0.5 cm, and the presence of thyroid capsular invasion (TCI) and multifocality) and three (maximum diameter of positive CLN (MDCLN) > 0.5 cm, the presence of multifocality and nodular goiter, iNG) factors were identified as independent risk factors for CLNM and LLNM, respectively. In bilateral PTMC, three (age < 55 and presence of TCI and multifocality in at least one side of thyroid lobe) and two (MDCLN > 0.5 cm and presence of nodular goiter (iNG)) factors were identified as independent factors for CLNM and LLNM, respectively. Predictive models of CLNM and LLNM for patients with unilateral disease and of CLNM for patients with the bilateral disease were established based on the described risk factors. Bilateral patients with positive CLNM were also stratified into different subgroups according to the presence and absence of independent risk factors. **Conclusion:** An evaluation system based on independent factors of CLNM and LLNM for PTMC patients with bilateral and unilateral disease was established. Our newly established evaluation system can efficaciously quantify risks of CLNM and LLNM for PTMC patients with bilateral and unilateral disease and may guide individual treatment strategy including both surgical and postoperative adjuvant treatment of the neck region for these patients.

## 1. Introduction

Papillary thyroid carcinoma (PTC) is the most prevalent type of thyroid cancer and has seen a continued increase in incidence over the recent decades globally [1,2]. PTC that has a maximum tumor diameter of 1.0 cm or less is defined as papillary thyroid microcarcinoma (PTMC), accounting for over half of all newly diagnosed thyroid cancer cases [3,4]. Although the nature of PTMC is generally indolent and the long-term prognosis is satisfactory, cervical lymph node involvement remains a concern [5], especially for those with bilateral lesions where lymph node metastasis rate may reach up to 60% [6]. Considering that the 2015 American Thyroid Association (ATA) places all intrathyroidal PTMCs, whether unifocal or multifocal, in the low-risk category, it is important to clarify the risk of lymph node metastasis involvement in PTMC [7]. It has also been reported that the occurrence of lymph node involvement and locoregional recurrence are both significantly more frequent in PTC patients with bilateral lesions than those with unilateral lesions [8]. In addition, the prevalence of BRAF V600E mutation in patients with bilateral PTMC was also significantly higher than in those with unilateral disease [9], indicating that marked differences exist between these two different entities. 

Whether prophylactic central lymph node dissection (CLND) should be conducted for PTMC patients with no clinically detected central lymph node metastasis (CLNM) is still controversial. Although the occult CLNM rate is not negligible with an incidence rate over 60% in some series [10,11], the relatively higher incidence of postoperative complications caused by CLND put into question the routine conduction of prophylactic surgery involving this region [12]. Developing an effective method to accurately predict cervical metastasis is thus crucial for the surgical management strategy of PTMC patients. Previous studies have revealed significantly different histopathological findings including lymph node metastasis between patients with bilateral and unilateral PTMC, however, none of these studies have quantitatively summarized the risk of cervical lymph node metastasis including both CLNM and lateral lymph node metastasis (LLNM) in PTMC patients with bilateral and unilateral diseases. Here in our current research, a comprehensive and meticulous evaluating system that can efficaciously quantify risks of CLNM and LLNM for bilateral and unilateral PTMC was established.

## 2. Materials and methods

### 2.1. Study Population

Between 2018 and 2020, 1477 patients with PTC received initial surgery at three clinical centers: Department of Otorhinolaryngology, Head and Neck Surgery at the Eye, Ear, Nose, and Throat Hospital of Fudan University, Department of General Surgery at Ruijin Hospital of Shanghai Jiao Tong University School of Medicine; and Department of General Surgery, Civil Aviation Shanghai Hospital. Among them, 751 patients were diagnosed with PTMC by postoperative pathology. Patients meeting any of the following conditions were excluded from our study: (1) having received thyroid-related surgery previously (*n* = 22); (2) history or coexistence of other primary tumors (*n* = 12). As a result, a total of 717 patients were enrolled for further analysis. This study was approved by the Institutional Ethics Committee of the Eye and ENT Hospital of Fudan University and the Ruijin Hospital of Shanghai Jiao Tong University School of Medicine. 

### 2.2. Surgical Management and Clinicopathological Features

Clinical and pathological features were retrospectively collected and analyzed. Preoperative fine-needle aspiration (FNA) was performed in all enrolled patients with PTMC and diagnosed by cytology. A total thyroidectomy or thyroid lobectomy was conducted for all patients in our cohort. Central lymph node dissection (CLND) was also performed for all patients considering both prophylactic and therapeutic purposes. Lateral lymph node dissection (LLND) was performed therapeutically for those with pre-operatively detected lateral lymph node metastasis (LLNM) using both preoperative ultrasonography and FNA. The LLND was performed for those with clinically detected lateral lymph nodes that were highly suspected as having tumor involvement using preoperative ultrasonography but later proven LLNM negative by FNA biopsy. Patients enrolled were treated with postoperative TSH suppression and radioactive iodine (RAI) therapy according to the 2015 American Thyroid Association Guidelines [7]. For patients receiving CLND only, if positive LLNM was found by ultrasonography and FNA within six months after initial surgery, they would be regarded as having lateral involvement at the time of operation. The thyroid glands were categorized into three equal volumes (upper portion, middle portion, and lower portion) based on the consensus of most clinical medical centers. Patients with tumors involving both sides of thyroid lobes were defined as having bilateral disease.

### 2.3. Statistical Analyses

The categorical and continuous variables were compared using the Pearson Chi-square test and independent t-test, respectively. Logistic univariate and multivariate regression analyses were used for screening out risk factors that were significantly correlated with LLNM by the SPSS 24.0 package (SPSS Inc., Chicago, IL, USA). A *p*-value of <0.05 was considered statistically significant. The prediction models for the risk of CLNM and LLNM were created based on the selected independent risk factors, respectively; and the corresponding concordance index (C-index), receiver operating characteristic (ROC) curve, and the calibration curve were constructed using R software (version 3.5.1; R Development Core Team, Bell Laboratories, Lucent Technologies, Murray Hill, NJ, USA).

## 3. Results

### 3.1. Clinicopathological Characteristics of PTMC Patients with Unilateral and Bilateral Diseases

A total of 582 (81.2%) patients were confirmed as having unilateral disease and were classified into the Uni group among all PTMC patients, while the other 135 (18.8%) patients that were diagnosed as having the bilateral disease were categorized into the Bil group. CLND was conducted routinely for all patients while LLND was only conducted for 47 (6.6%) patients with clinically detected or highly suspected LLNM. As a result, 307 (42.8%) and 41 (5.7%) patients were confirmed as having CLNM and LLNM by postoperative pathology. In addition, eight patients receiving CLND alone and detected as having lateral neck involvement within six months in post-operation follow-up were also regarded as having preoperative LLNM. In total, 49 (6.8%) patients were considered as having LLNM at the time of initial operation.

The basic clinicopathological features of patients within the Uni and Bil groups are shown and compared in Table 1. Tumor size was significantly larger in patients of the Bil group than that of the Uni group ((0.58 ± 0.23) cm vs. (0.54 ± 0.22) cm, *p*-value = 0.031). The presence of thyroid capsular invasion (TCI, 40.7% vs. 27.7%, *p*-value = 0.003), multifocality in at least one side of thyroid lobe (49.6% vs. 23.2%, *p*-value = 0.000), upper portion tumor of thyroid (33.3% vs. 24.4%, *p*-value = 0.033), and Hashimoto thyroiditis (HT, 28.1% vs. 18.4%, *p*-value = 0.011) were significantly more frequent in patients of the Bil group than those of the Uni group. In terms of cervical lymph node metastasis, the overall CLNM and LLNM rates were 42.8% (307 in 717) and 6.8% (49 in 717), respectively, for all patients enrolled, and the incidence of CLNM was significantly higher in patients within the Bil group than those within the Uni group (55.6% vs. 39.9%, *p*-value = 0.001). Detailed information on CLNM was also collected and analyzed, and the result showed that although patients with bilateral and unilateral disease exhibited comparable levels in terms of counts of positive central lymph node (CLN), patients within the Bil group showed significantly larger positive CLN sizes than those within the Uni group ((0.64 ± 0.42) cm vs. (0.49 ± 0.36) cm, *p*-value = 0.003). Moreover, for patients with positive CLNM, those with bilateral disease also showed a significantly higher lateral neck involvement rate (25.3% vs. 12.9%, *p*-value = 0.011).

### 3.2. Comparisons between PTMC with or without CLNM and LLNM for Patients within Uni and Bil Groups

Further analyses were conducted between patients with or without CLNM and LLNM within Uni and Bil groups (Shown in Table 2). For patients in the Uni group, the age of patients with positive CLNM was significantly younger than those with negative CLNM (40.32 ± 11.98 years old vs. 45.20 ± 12.26 years old, *p*-value = 0.000). The maximum tumor diameter (MTD) was larger in patients with positive CLNM than in those with negative CLNM (*p*-value = 0.000). In addition, being male and the presence of TCI and ipsilateral multifocality were significantly more frequent in patients with positive CLNM (*p*-value = 0.000, 0.000, and 0.000, respectively). We further divided patients with positive CLNM into two subgroups according to the status of lateral lymph node involvement: patients with positive (*n* = 30) and negative (*n* = 202) LLNM. The presence of ipsilateral multifocality (70.0% vs. 35.1%) and nodular goiter (iNG, 56.7% vs. 23.8%) were significantly more common in patients with positive LLNM than those without (*p*-value = 0.000 and 0.000, respectively). Patients with positive LLNM also showed significantly larger sizes and higher counts of positive CLN than those with negative LLNM (*p*-value = 0.000 and 0.001, respectively).

For patients in the Bil group, younger age, larger MTD, being male, and a more common presence of TCI and multifocality in at least one side of the thyroid lobe were also found in patients with positive CLNM compared with those with negative CLNM (Shown in Table 2). However, the presence of iNG, which was proven to be significantly more frequent in patients with positive CLNM among all patients within the Uni group, showed no difference in patients with positive CLNM among those within the Bil group (28.0% vs. 28.3%, *p*-value = 0.966). For patients with positive CLNM within the Bil group, the percentages of factors including male patients and iNG were significantly higher in patients with positive LLNM (*p*-value = 0.021 and 0.006, respectively). Additionally, patients with positive LLNM also showed a larger size of positive CLN than those with negative LLNM (*p*-value = 0.000).

### 3.3. Creation of Risk Prediction Model for Cervical Lymph Node Metastasis of PTMC Patients within Uni Group

The result of the univariate and multivariate regression analyses showed that five factors (age less than 55 years old, male, MTD ≥ 0.5cm, and the presence of TCI and multifocality) were proven to be independent risk factors of CLNM for PTMC patients within the Uni group (Shown in Table 3). Meanwhile, for patients with positive CLNM, the result of multivariate analysis exhibited that three factors (maximum diameter of positive CLN (MDCLN) > 0.5cm, the presence of multifocality, and iNG) were screened out as independent risk factors of LLNM. The prediction models for quantitatively assessing the risk of CLNM and LLNM for all patients within the Uni group and unilateral patients with positive CLNM respectively were then created based on the independent risk factors described (Shown in Figure 1A,B). To validate the accuracy of the newly created nomogram, an internal validation by 1000 bootstrap resamples was performed and assessed in terms of the C-index. Validation results returned a C-index of 0.806 (95% CI, 0.769–0.843), and 0.803 (95% CI, 0.790–0.816) after bootstrapping, demonstrating our nomogram’s excellent accuracy in CLNM risk prediction. The ROC curve and the calibration plot are shown in Figure 1C,E, both exhibiting satisfactory agreement between the actual and predicted probability of CLNM for patients in the Uni group. For the nomogram used to assess the risk of LLNM in unilateral patients with positive CLNM, a C-index of 0.938 (95% CI, 0.881–0.994) was yielded, and 0.931 (95% CI, 0.913–0.949) after bootstrapping. The ROC curve and the calibration plot (Shown in Figure 1D,F) also confirmed high efficiency and accuracy for predicting LLNM.

### 3.4. Creation of Risk Prediction Model for Cervical Lymph Node Metastasis of PTMC Patients within the Bil Group

Three factors including age less than 55 and the presence of TCI and multifocality in at least one side of the thyroid lobe were confirmed as independent risk factors of CLNM for patients within the Bil group by multivariate analysis (shown in Table 4). Similarly, the prediction model for quantitatively assessing the risk of CLNM for these patients was established (shown in Figure 2A), and the ROC and the calibration curves were plotted and shown in Figure 2B,C, both indicating a high degree of accuracy of our newly created model (with C-index of 0.783 (95% CI, 0.706–0.860) and 0.776 (95% CI, 0.754–0.798) for training group and after bootstrapping respectively). 

For bilateral PTMC patients with CLNM, two factors including MDCLN > 0.5 cm and the presence of iNG were screened out as independent factors of LLNM. These patients were then divided into four subgroups according to the presence of the two factors: patients with none of the two risk factors, patients with iNG only, patients with MDCLN > 0.5 cm only, and patients with both of the two risk factors. The incidences of LLNM for patients within different subgroups were shown and compared in Table 5. Patients exhibiting both of the two risk factors (8 in 11, 72.7%) showed a much higher LLNM rate than those within the other three subgroups, while only 1 (2.9%) in 35 patients with none of the two factors was proven to have lateral neck involvement in our cohort.

### 3.5. Risk Stratification and Cervical Lymph Node Metastasis Risk Assessment Flow Chart for PTMC Patients

Each factor enrolled for the construction of the nomogram has its own risk points. Patients within different groups would gain a total risk score by summing up the risk scores of each factor based on their own nomogram. According to the distribution of the total risk scores, all patients within the Uni group, unilateral patients with positive CLNM, and patients within the Bil group were separately classified into different subgroups with significantly distinct CLNM or LLNM risks (shown in Table 5, *p*-value = 0.000, 0.000, and 0.000, respectively) by different cutoff values.

The risk stratification according to the nomogram score was shown in Table 6. The aforementioned three nomograms and the risk stratification strategy of LLNM for bilateral patients with positive CLNM were further integrated and were presented as a comprehensive flow diagram for quantitatively evaluating the risk of cervical lymph node involvement for all patients with PTMC (exhibited in Figure 3).

## 4. Discussion

Previous studies have demonstrated that bilateral PTCs usually present more aggressively, and more advanced, as well as having poorer recurrence and survival outcomes [13]. Bilateral disease was also reported to be significantly associated with clinicopathological features including both primary tumor sites such as larger tumor size and thyroid capsular invasion, and cervical lymph node metastasis [14]. Furthermore, here in our study, compared with the previous research, more detailed factors have been enrolled and analyzed. As a result, the presence of multifocality in at least one side of the thyroid lobe, upper portion tumor of the thyroid, and iNG were significantly more frequent in patients with bilateral disease. In terms of further information regarding cervical lymph node metastasis, central lymph node metastasis (CLNM) was significantly more common in patients within the Bil group than those within the Uni group, where higher counts and larger-sized central lymph nodes were found in patients within the Bil group. For patients with positive CLNM, those within the Bil group also showed a significantly higher incidence of LLNM. All these differences further indicate the extremely more invasive nature of bilateral tumors. Wang et al. [15] proved that bilateral tumors within one patient share the same clone source, and this result also suggests that bilateral disease may be a more aggressive status of intra-thyroid metastasis from the primary tumor at the contralateral lobe, which further validates our conclusions. The aforementioned significant difference between PTMC patients with unilateral and bilateral diseases further indicates the significant implications of our study to discuss patients within these two groups separately.

Several studies have revealed that male gender, larger tumor size, multifocality, younger age, and the presence of TCI are independent risk factors of CLNM for patients with PTMC [16,17,18], however, little research has focused on the central neck involvement for PTMC patients with unilateral and bilateral diseases respectively. Here in our research, factors including age < 55 years old, male, MTD ≥ 0.5cm, and the presence of TCI and multifocality were identified as independent factors of CLNM for PTMC patients with unilateral disease, while these factors including age < 55 years old, and the presence of TCI and multifocality for those with bilateral disease. Two prediction models for quantitatively assessing the risk of CLNM for PTMC patients with unilateral and bilateral diseases were established based on their respective independent risk factors. Then, patients in each group were stratified into three subgroups with significantly different CLNM risks according to the distribution of the total score received from their own prediction models. For PTMC patients with unilateral disease, who are traditionally considered as having a low risk of CLNM among all PTC patients, a subgroup of patients were screened out and proven to have a much higher risk of CLNM (140 in 173, 80.9%). However, for PTMC patients with bilateral disease, who displayed a significantly higher incidence of CLNM than those with unilateral disease, a small subgroup of patients with no incidence of CLNM has also been selected (0 in 14, 0.0%), showing much lower CLNM risk compared to those classified as a high-risk subgroup (66 in 90, 73.3%). For patients with PTMC, aside from resection of the primary tumor site, a “wait and see” strategy for clinical negative central neck region is recommended by many guidelines including the 2015 American Thyroid Association (ATA) guideline to avoid unnecessary surgery-related complications. However, several research works have revealed that the incidence of central lymph node involvement for patients with PTMC is not that low, with reported CLNM ranging from 27.4% to 53.9% [19,20,21], and that cervical lymph node involvement is likely associated with an elevated incidence of loco-regional tumor recurrence [22]. Considering this, an effective method that can accurately assess the risk of CLNM for PTMC patients is necessary. According to our results, a more cautious examination of central neck regions, as well as a more frequent postoperative follow-up, should be conducted for unilateral and bilateral patients with a total score of no less than 180 and 90, according to their respective nomograms. Prophylactic CLND could also be considered as a second choice given the extremely high CLNM risk. However, for those within the low CLNM risk subgroup, resection of the primary tumor site is enough, and intervention of the central neck region is unnecessary. 

Existing literature showed that factors such as upper portion tumor and a high count of positive central lymph nodes are independent risk factors of LLNM for PTMC patients [23,24]. Here in our research, the risk of LLNM was also quantitatively analyzed for PTMC patients within the Uni group. The LLNM risk for patients in low LLNM risk subgroup was only 2.0% yet could reach up to 72.2% for those in the high LLNM risk subgroup. For those in the Bil group, only two factors including the presence of iNG and the maximum diameter of positive lymph nodes in the central compartment >0.5 cm were confirmed as independent factors, and those with none of these two factors showed an extremely low LLNM risk. Given that those exhibiting both of these two risk factors showed a high LLNM risk of 72.7%, the rationality and validity of our classification are confirmed. For unilateral and bilateral patients within high LLNM risk groups according to their respective prediction models, a more frequent postoperative follow-up schedule is necessary. For those defined as having low LLNM risk, therapeutic CLND is enough and no additional intervention involving the lateral neck is needed.

## 5. Conclusion

An evaluation system based on independent factors of CLNM and LLNM for PTMC patients with bilateral and unilateral disease was established. Our newly established evaluating system can efficaciously quantify risks of CLNM and LLNM for PTMC patients with bilateral and unilateral disease and may guide individual treatment strategy including both surgical and postoperative adjuvant treatment of neck region for these patients.

## Figures and Tables

**Figure 1 jcm-11-04929-f001:**
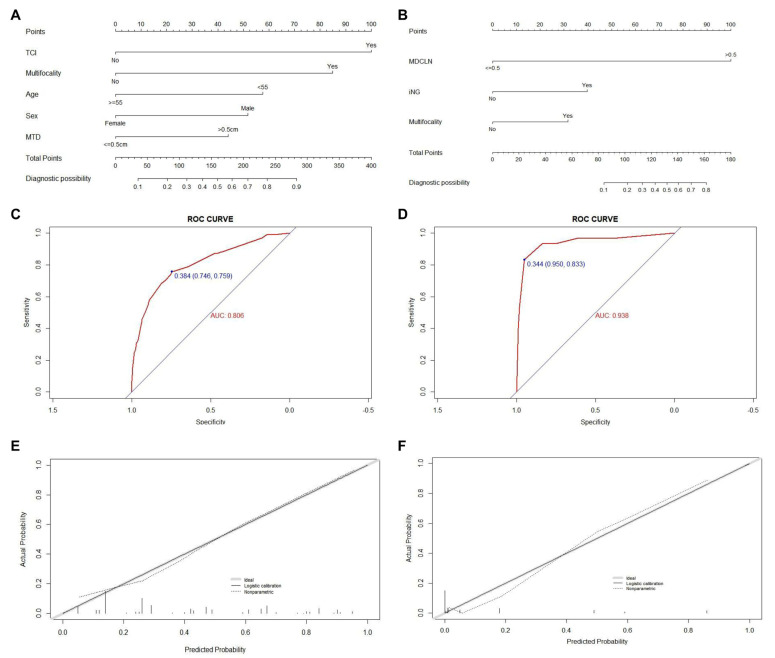
Construction, assessment, and validation of the predictive model of CLNM and LLNM. (**A**,**B**) The nomograms for predicting CLNM and LLNM risk in PTMC patients within the Uni group, respectively; (**C**,**D**) the ROC curve and AUC of the nomograms for predicting CLNM and LLNM risk in PTMC patients within the Uni group, respectively; (**E**,**F**) the calibration curves of the nomogram for predicting CLNM and LLNM risk in PTMC patients within the Uni group, respectively. Actual probability is plotted on the y-axis, and nomogram predicted probability on the x-axis. PTMC, papillary thyroid microcarcinoma; CLNM, central lymph node metastasis; LLNM, lateral lymph node metastases; TCI, thyroid capsular invasion; MTD, maximum tumor diameter; MDCLN, the maximum diameter of positive central lymph node; iNG, ipsilateral nodular goiter; ROC, receiver operating characteristics.

**Figure 2 jcm-11-04929-f002:**
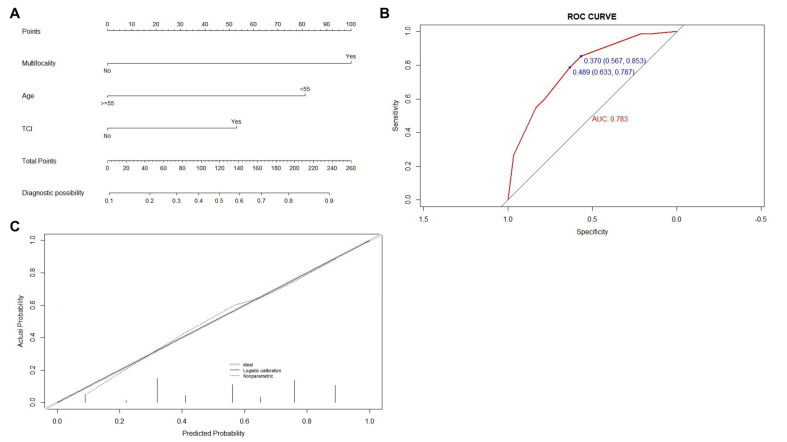
Construction, assessment, and validation of the predictive model of CLNM for PTMC patients within the Bil group. (**A**) The nomograms for predicting CLNM risk in PTMC patients within the Bil group; (**B**) the ROC curve and AUC of the nomogram for predicting CLNM risk in PTMC patients within the Bil group; (**C**) the calibration curve of the nomogram for predicting CLNM risk in PTMC patients within the Bil group. Actual probability is plotted on the y-axis, and nomogram predicted probability on the x-axis. PTMC, papillary thyroid microcarcinoma; CLNM, central lymph node metastasis; TCI, thyroid capsular invasion; ROC, receiver operating characteristics.

**Figure 3 jcm-11-04929-f003:**
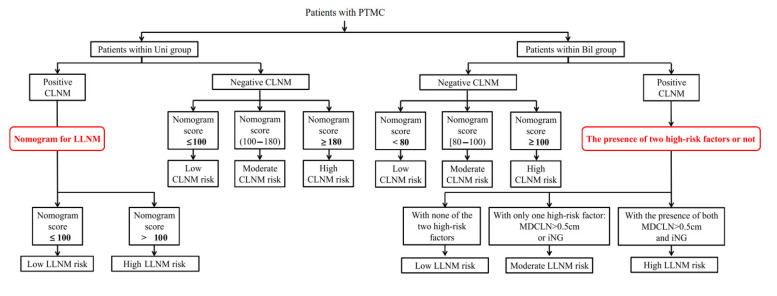
A flow chart of cervical lymph node metastasis risk including both CLNM and LLNM for PTMC patients with unilateral and bilateral diseases. CLNM, central lymph node metastasis; LLNM, lateral lymph node metastases; PTMC, papillary thyroid microcarcinoma; iNG, ipsilateral nodular goiter.

**Table 1 jcm-11-04929-t001:** The clinicopathological characteristics of patients with PTMC.

	All Patients		Unilateral		Bilateral		
	*n* = 717	%	*n* = 582	%	*n* = 135	%	*p* Value
**Age (mean ± SD)**	43.40 ± 12.31		43.25 ± 12.37		44.04 ± 12.09		0.502
**BMI (mean ± SD)**	23.60 ± 3.68		23.51 ± 3.66		23.98 ± 3.77		0.180
**Maximum tumor diameter (mean ± SD)**	0.55 ± 0.23		0.54 ± 0.22		0.58 ± 0.23		0.031
**Gender**							0.499
Male	230	32.1	190	32.6	40	29.6	
Female	487	67.9	392	67.4	95	70.4	
**Thyroid capsular invasion**							0.003
No	501	69.9	421	72.3	80	59.3	
Yes	216	30.1	161	27.7	55	40.7	
**Multifocality**							0.000
Absent	515	71.8	447	76.8	68	50.4	
Present	202	28.2	135	23.2	67	49.6	
**Tumor location**							0.033
Upper portion	187	26.1	142	24.4	45	33.3	
Middle/Lower portion	530	73.9	440	75.6	90	66.7	
**PTMC with Hashimoto thyroiditis**							0.011
No	572	79.8	475	81.6	97	71.9	
Yes	145	20.2	107	18.4	38	28.1	
**PTMC with ipsilateral nodular goiter**							0.942
No	517	72.1	420	72.2	97	71.9	
Yes	200	27.9	162	27.8	38	28.1	
**CLNM**							0.001
No	410	57.2	350	60.1	60	44.4	
Yes	307	42.8	232	39.9	75	55.6	
**Number of positive CLN**							0.621
(For patients with CLNM only, *n* = 307)							
1–2	178	58.0	136	58.6	42	56.0	
3–4	71	23.1	55	23.7	16	21.3	
≥5	58	18.9	41	17.7	17	22.7	
**Maximum diameter of positive CLN**							0.003
Mean ± SD, cm	0.53 ± 0.38		0.49 ± 0.36		0.64 ± 0.42		
Median (range), cm	0.4 (0.1–2.5)		0.4 (0.1–2.5)		0.5 (0.1–2.0)		
							0.024
≤0.5cm	216	70.4	171	73.7	45	60.0	
>0.5cm	91	29.6	61	26.3	30	40.0	
**LLNM**							0.011
No	258	84.0	202	87.1	56	74.7	
Yes	49	16.0	30	12.9	19	25.3	

SD, standard error; PTMC, papillary thyroid microcarcinoma; BMI, body mass index; CLNM, central lymph node metastasis; CLN, central lymph node; LLNM, lateral lymph node metastasis.

**Table 2 jcm-11-04929-t002:** The clinicopathological characteristics of PTMC patients with different lymph node metastasis status within Bil and Uni groups.

	Uni Group (*n* = 582)						Bil Group (*n* = 135)					
	All Patients (*n* = 582) (*n* (%))		*p* Value	Patients with CLNM (*n* = 232) (*n* (%))		*p* Value	All Patients (*n* = 135) (*n* (%))		*p* Value	Patients with CLNM (*n* = 75) (*n* (%))		*p* Value
	No-CLNM	CLNM		No-LLNM	LLNM		No-CLNM	CLNM		No-LLNM	LLNM	
	*n* = 350	*n* = 232		*n* = 202	*n* =30		*n* = 60	*n* = 75		*n* = 56	*n* = 19	
**Age (mean ± SD)**	45.20 ± 12.26	40.32 ± 11.98	0.000	40.78 ± 12.15	37.23 ± 10.43	0.130	48.82 ± 11.74	40.23 ± 11.03	0.000	39.82 ± 10.71	41.42 ± 12.16	0.588
**BMI (mean ± SD)**	23.44 ± 3.32	23.61 ± 4.13	0.588	23.66 ± 4.18	23.30 ± 3.82	0.658	23.25 ± 3.33	24.56 ± 4.01	0.045	24.46 ± 4.22	24.85 ± 3.40	0.720
**Maximum tumor diameter (mean ± SD)**	0.49 ± 0.22	0.61 ± 0.23	0.000	0.61 ± 0.20	0.63 ± 0.21	0.654	0.53 ± 0.23	0.63 ± 0.23	0.009	0.61 ± 0.22	0.68 ± 0.26	0.285
**Gender**			0.000			0.749			0.070			0.021
Male	88 (25.1)	102 (44.0)		88 (43.6)	14 (46.7)		13 (21.7)	27 (36.0)		16 (28.6)	11 (57.9)	
Female	262 (74.9)	130 (56.0)		114 (56.4)	16 (53.3)		47 (78.3)	48 (64.0)		40 (71.4)	8 (42.1)	
**Thyroid capsular invasion**			0.000			0.202			0.009			0.739
No	303 (86.6)	118 (50.9)		106 (52.5)	12 (40.0)		43 (71.7)	37 (49.3)		27 (48.2)	10 (52.6)	
Yes	47 (13.4)	114 (49.1)		96 (47.5)	18 (60.0)		17 (28.3)	38 (50.7)		29 (51.8)	9 (47.4)	
**Multifocality**			0.000			0.000			0.000			0.060
Absent	307 (87.7)	140 (60.3)		131 (64.9)	9 (30.0)		43 (71.7)	25 (33.3)		22 (39.3)	3 (15.8)	
Present	43 (12.3)	92 (39.7)		71 (35.1)	21 (70.0)		17 (28.3)	50 (66.7)		34 (60.7)	16 (84.2)	
**Tumor location**			0.752			0.683			0.142			0.435
Upper portion	87 (24.9)	55 (23.7)		47 (23.3)	8 (26.7)		24 (40.0)	54 (72.0)		17 (30.4)	4 (21.1)	
Middle/Lower portion	263 (75.1)	177 (76.3)		155 (76.7)	22 (73.3)		36 (60.0)	21 (28.0)		39 (69.6)	15 (78.9)	
**Number of positive CLN**			/			0.001			/			0.209
1–2	/	136 (58.6)		126 (62.4)	10 (33.3)		/	42 (56.0)		34 (60.7)	8 (42.1)	
3–4	/	55 (23.7)		47 (23.3)	8 (26.7)		/	16 (21.3)		12 (21.4)	4 (21.1)	
≥5	/	41 (17.7)		29 (14.4)	12 (40.0)		/	17 (22.7)		10 (17.9)	7 (36.8)	
**Maximum diameter of positive CLN**			/			0.000			/			0.000
≤0.5cm	/	171 (73.7)		169 (83.7)	2 (6.7)		/	45 (60.0)		42 (75.0)	3 (15.8)	
>0.5cm	/	61 (26.3)		33 (16.3)	28 (93.3)		/	30 (40.0)		14 (25.0)	16 (84.2)	
**PTMC with Hashimoto thyroiditis**			0.309			0.964			0.467			0.634
No	281 (80.3)	194 (83.6)		169 (83.7)	25 (83.3)		45 (75.0)	52 (69.3)		38 (67.9)	14 (73.7)	
Yes	69 (19.7)	38 (16.4)		33 (16.3)	5 (16.7)		15 (25.0)	23 (30.7)		18 (32.1)	5 (26.3)	
**PTMC with ipsilateral nodular goiter**			0.936			0.000			0.966			0.006
No	253 (72.3)	167 (72.0)		154 (76.2)	13 (43.3)		43 (71.7)	54 (72.0)		45 (80.4)	9 (47.4)	
Yes	97 (27.7)	65 (28.0)		48 (23.8)	17 (56.7)		17 (28.3)	21 (28.0)		11 (19.6)	10 (52.6)	

PTMC, papillary thyroid microcarcinoma; BMI, body mass index; CLNM, central lymph node metastasis; CLN, central lymph node; LLNM, lateral lymph node metastasis.

**Table 3 jcm-11-04929-t003:** Univariate and multivariate analyses for PTMC patients with unilateral disease.

	Univariate Analysis		Multivariate Analysis			Univariate Analysis		Multivariate Analysis	
	Hazard Ratio (95% CI)	*p* Value	Hazard Ratio (95% CI)	*p* Value		Hazard Ratio (95% CI)	*p* Value	Hazard Ratio (95% CI)	*p* Value
** *Factors selected* **					** *Factors selected* **				
**Age**		** *0.000* **		**0.000**	**Age**		0.261		
≥55 vs. <55	***0.426 (0.273*–*0.666)***		***0.360 (0.213*–*0.609)***		≥55 vs. <55	0.426 (0.096–1.886)			
**BMI**		0.436			**BMI**		0.596		
>23 vs. ≤23	0.876 (0.629–1.222)				>23 vs. ≥23	0.812 (0.375–1.758)			
**Gender**		** *0.000* **		** *0.000* **	**Gender**		0.749		
Male vs. Female	***2.336 (1.639*–*3.329)***		***2.505 (1.643*–*3.818)***		Male vs. Female	1.134 (0.525–2.446)			
**TCI**		** *0.000* **		** *0.000* **	**TCI**		0.205		
Yes vs. No	***6.228 (4.171*–*9.299)***		***5.894 (3.736*–*9.299)***		Yes vs. No	1.656 (0.759–3.616)			
**Maximum tumor diameter**		** *0.000* **		** *0.000* **	**Maximum tumor diameter**		0.228		
>0.5 cm vs. ≤0.5 cm	***2.788 (1.978*–*3.929)***		***2.181 (1.456*–*3.269)***		>0.5cm vs. ≤0.5cm	1.694 (0.719–3.993)			
**Tumor location**		0.752			**Tumor location**		0.683		
Upper vs. Middle/Lower	0.939 (0.637–1.384)				Upper vs. Middle/Lower	1.199 (0.501–2.870)			
**Multifocality**		** *0.000* **		** *0.000* **	**Multifocality**		** *0.001* **		** *0.010* **
Yes vs. No	***4.692 (3.103*–*7.095)***		***4.514 (2.831*–*7.196)***		Yes vs. No	***4.305 (1.872*–*9.899)***		***4.439 (1.423*–*13.847)***	
**PTMC with ipsilateral nodular goiter**		0.936			**PTMC with ipsilateral nodular goiter**		** *0.000* **		** *0.003* **
Yes vs. No	1.015 (0.701–1.470)				Yes vs. No	***4.196 (1.901*–*9.258)***		***6.311 (1.883*–*21.159)***	
**PTMC with Hashimoto thyroiditis**		0.310			**PTMC with Hashimoto thyroiditis**		0.964		
Yes vs. No	0.798 (0.516–1.234)				Yes vs. No	1.024 (0.366–2.869)			
					**Maximum diameter of positive CLN**		** *0.000* **		** *0.000* **
					>0.5cm vs. ≤0.5cm	***71.697 (16.284*–*315.670)***		***107.399 (20.011*–*576.400)***	
					**Number of positive CLN**		** *0.001* **		0.826
					≥3 vs. <3	***2.282 (1.429*–*3.644)***		0.924 (0.458–1.866)	

CI, confidence interval; PTMC, papillary thyroid microcarcinoma; TCI, thyroid capsular invasion; BMI, body mass index; CLN, central lymph node.

**Table 4 jcm-11-04929-t004:** Univariate and multivariate analyses for PTMC patients with bilateral disease.

	Univariate Analysis		Multivariate Analysis			Univariate Analysis		Multivariate Analysis	
	Hazard Ratio (95% CI)	*p* Value	Hazard Ratio (95% CI)	*p* Value		Hazard Ratio (95% CI)	*p* Value	Hazard Ratio (95% CI)	*p* Value
** *Factors selected* **					** *Factors selected* **				
**Age**		** *0.007* **		** *0.004* **	**Age**		0.716		
≥55 vs. <55	** *0.308 (0.131–0.724)* **		** *0.224 (0.080–0.626)* **		≥55 vs. <55	1.312 (0.303–5.683)			
**BMI**		0.609			**BMI**		0.255		
>23 vs. ≤23	1.199 (0.599–2.401)				>23 vs. ≤23	1.952 (0.617–6.173)			
**Gender**		0.072			**Gender**		* **0.025** *		0.293
Male vs. Female	2.034 (0.938–4.411)				Male vs. Female	** *3.437 (1.168–10.118* ** *)*		2.052 (0.538–7.824)	
**TCI**		** *0.009* **		** *0.020* **	**TCI**		0.739		
Yes vs. No	** *2.598 (1.263–5.344)* **		** *2.624 (1.161–5.929)* **		Yes vs. No	0.838 (0.296–2.375)			
**Maximum tumor diameter**		** *0.038* **		0.133	**Maximum tumor diameter**		0.850		
>0.5 cm vs. ≤0.5 cm	** *2.074 (1.040–4.137)* **		1.836 (0.832–4.055)		>0.5 cm vs. ≤0.5 cm	1.109 (0.378–3.251)			
**Tumor location**		0.143			**Tumor location**		0.438		
Upper vs. Middle/Lower	0.583 (0.283–1.200)				Upper vs. Middle/Lower	0.612 (0.177–2.117)			
**Multifocality**		** *0.000* **		** *0.000* **	**Multifocality**		0.071		
Yes vs. No	** *5.059 (2.417–10.590)* **		** *6.278 (2.728–14.451)* **		Yes vs. No	3.451 (0.899–13.241)			
**PTMC with ipsilateral nodular goiter**		0.966			**PTMC with ipsilateral nodular goiter**		** *0.008* **		** *0.038* **
Yes vs. No	0.984 (0.463–2.092)				Yes vs. No	** *4.545 (1.489–13.876)* **		** *4.375 (1.083–17.670)* **	
**PTMC with Hashimoto thyroiditis**		0.468			**PTMC with Hashimoto thyroiditis**		0.635		
Yes vs. No	1.327 (0.619–2.846)				Yes vs. No	0.754 (0.235–2.417)			
					**Maximum diameter of positive CLN**		** *0.000* **		** *0.000* **
					>0.5 cm vs. ≤0.5 cm	** *16.000 (4.052–63.185)* **		** *13.868 (3.226–59.610)* **	
					**Number of positive CLN**		0.090		
					≥3 vs. <3	1.707 (0.919–3.170)			

**Table 5 jcm-11-04929-t005:** Risk stratification of CLNM and LLNM for PTMC patients within Uni and Bil groups.

			Low Risk (TS ≤ 100)	Moderate Risk (100 < TS < 180)	High Risk (TS ≥ 180)	
			(*n* = 197, %)	(*n* = 212, %)	(*n* = 173, %)	*p* Value
**Uni group**	**ALL patients (*n* = 582)**	**Negative CLNM**	173 (87.8)	144 (67.9)	33 (19.1)	0.000
		**Positive CLNM**	24 (**12.2**)	68 (**32.1**)	140 (**80.9**)	
			** Low risk (TS ≤ 100)**		**High risk (TS > 100)**	
			**(*n* = 196, %)**		**(*n* = 36, %)**	***p* value**
	**Patients with positive CLNM (*n* = 232)**	**Negative LLNM**	192 (98.0)		10 (27.8)	0.000
		**Positive LLNM**	4 (**2.0**)		26 (**72.2**)	
			**Low risk (TS < 80)**	** Moderate risk (80 ≤ TS < 100) **	** High risk (TS ≥ 100) **	
			** (*n* = 14, %) **	** (*n* = 31, %) **	** (*n* = 90, %) **	***p* value**
**Bil group**	**All patients (*n* = 135)**	**Negative LLNM**	14 (100.0)	22 (71.0)	24 (26.7)	0.000
		**Positive LLNM**	0 (**0.0**)	9 (**29.0**)	66 (**73.3**)	
			**No risk factor**	** iNG only MDCLN > 0.5 cm only **	**Both two risk factors**	
			** (*n* = 35, %) **	** (*n* = 10, %) (*n* = 19, %) **	** (*n* = 11, %) **	***p* value**
	**Patients with positive CLNM (*n* = 75)**	**Negative LLNM**	34 (97.1)	8 (80.0) 11 (57.9)	3 (27.3)	0.000
		**Positive LLNM**	1 (**2.9**)	2 (**20.0**) 8 (**42.1**)	8 (**72.7**)	

PTMC, papillary thyroid microcarcinoma; CLNM, central lymph node metastasis; LLNM, lateral lymph node metastasis; iNG, ipsilateral nodular goiter; MDCLN, maximum diameter of positive central lymph node.

**Table 6 jcm-11-04929-t006:** Risk stratification of Uni and Bil group of PTC patients based on the model database.

	Uni Group CLNM Risk			Uni Group LLNM Risk		Bil Group CLNM Risk		
	Low	Moderate	High	Low	High	Low	Moderate	High
**Nomogramscore**	**0–100**	**100–180**	**>180**	**<100**	**>100**	**<80**	**80–100**	**>100**
**Value**	24/197	68/212	140/173	4/196	26/36	0/14	9/31	66/90
**%**	12.2	32.1	80.9	2	72.2	0	29	73.3

## Data Availability

The datasets generated and analyzed during the current study are available from the corresponding author on reasonable request.

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
