# Peer review of "Cervical Lymph Node Metastasis Differences in Patients with Unilateral or Bilateral Papillary Thyroid Microcarcinoma: A Multi-Center Analysis"

_jcm, 2022, doi:10.3390/jcm11164929_

Round 1

Reviewer 1 Report

Papillary thyroid microcarcinoma is currently the most frequent endocrine cancer. As the authors mentioned, the current classification guidelines do not comprehend the whole disease complexity. Regardless of how comprehensive the content of the guidelines is, certain therapeutic settings are still limited. 

 Major considerations:

1.     It would be important if the authors underlines the lack of information regarding PTMC.

Example:

§  The Updated AJCC/TNM (American Joint Committee on Cancer/Union for International Cancer) Staging System for Differentiated and Anaplastic Thyroid Cancer (10th edition) defines primary tumor’s category only by the size of the greatest dimension. 

§  The 2015 American Thyroid Association (ATA) places all intrathyroidal PTMCs, whether unifocal or multifocal, in the low-risk category. Only multifocal PTMCs with extrathyroidal extension (ETE) are considered to be in the intermediate-risk group. 

§  No recommendations regarding PTMC treatment in National Comprehensive Cancer Network (NCCN) 2022, European Thyroid Association (ETA) 2019 , and European Society for Medical Oncology (ESMO) 2019.

2.     In JCM 2021 there is an already published similar study, that the authors should analyze.  All data in this filed will contribute to a larger database that will be important for future guidelines. HîÈ›u L et al.Total Tumor Diameter and Unilateral Multifocality as Independent Predictor Factors for Metastatic Papillary Thyroid Microcarcinoma. J Clin Med. 2021 Aug 20;10(16):3707. doi: 10.3390/jcm10163707. PMID: 34442001; PMCID: PMC8396836.

3.     Please insert proper citations (ex. pg 2 line 81 is missing the citation)

4.     It is confusing to have ipsilateral Hashimoto thyroiditis (HT). Please consider to rephrase. HT is by default a diffuse autoimmune disease of the whole thyroid gland. To mention ipsilateral HT, someone will assume that the other lobe is not affected, which is not a practical situation.

5.     Please correct Figure 3. I supposed that the authors would like to conclude high LLNM risk, instead of CLNM, respectively LLNM risk not CLNM risk !!!!

6.     Also, please consider to be more clear about the scors.

7.     The conclusion needs to be improved and rephrased stating the clear results obtained in this study. 

“A comprehensive and meticulous evaluating system that can efficaciously quantify risks of neck lymph node metastasis including both CLNM and LLNM for PTMC patients with bilateral and unilateral disease respectively was established in our research”

Why your evaluation system is comprehensive and meticulous? It is not demonstrated in  the present study, so it would be better to skip. 

Minor considerations

1.     Please consider to adjust properly colums and rows in the tables, because the numbers are not displayed properly.

Author Response

Firstly, thank you very much for your precious time to review this paper and the meaningful and valuable comments. We have carefully read your suggestions and made the following modifications.

  1. According to your suggestion, we have optimized the background introduction of PTMC.
  2. According  to your suggestion, we have reorganized the references, and cite the referrence (doi: 10.3390/jcm10163707) in the discussion part.

  3.  

    Same as above.
  4. We modify all ipsilateral Hashimoto Thyroiditis to Hashimoto Thyroiditis in the revised article.

  5.  

    We have revised FIG3 and apologize for the previous labeling error. The prediction models of CLNM and LLNM are the research contents of this paper. According to Fig1 and Fig2, the final recommendation model of FIG3 is obtained

  6.  

    The risk stratification according to the nomogram score is shown in a new table, Table 6.

  7.  

    The conclusion of this article has been improved.

  8.  

     We have simplified the tables in the article.

Reviewer 2 Report

The authors' objective is to quantitatively predict the risk of neck lymph node metastasis for unilateral and bilateral papillary thyroid microcarcinomas.

The manuscript is well written and I think the manuscript is very interesting but it needs some clarifications.

Specific Comments:

1.     In the abstract line 18 “iNG” is the first time so change the acronym (not at line 21)

2.     In results, line 103 you write that 41 (5.7%) patients were confirmed as having  LLNM by postoperative pathology but at line 102 you write 47 (6.6%). Did the perform FNA and thyroglobulin wash of these lymph nodes?

3.     Please simplify all tables  (or change in horizontal table)

4.     Do you have the size of lateral lymph node?

5.     Do you have clinical data (first scintigraphy post I-131?, Thyroglobulin value? Neck ultrasound during the follow-up?

Author Response

  1. In the revised manuscript, we've changed the acronym of “iNG”.
  2. This is because for lateral lymph nodes with high suspicion of metastasis indicated by preoperative examination, prophylactic modified LLND was proformed even if the preoperative FNA did not give a clear diagnosis, which was based on the risk of lymph nodes metastasis and the patient's willingness to operate.

  3. We have simplified the tables in the article.

  4. The mean maximum diameter of lateral metastatic lymph nodes was 1.16cm, while the median was 1.2cm.

  5. This article is only a retrospective study to predict the risk of neck lymph node metastasis for unilateral and bilateral papillary thyroid microcarcinomas (PTMC), lacking systematic follow-up data, including First Scintigraphy Post I-131, Thyroglobulin value, and Neck ultrasound etc. According to your suggestion, we will carry out related studies on prognosis in future research.

Round 2

Reviewer 1 Report

Minor considerations

1.     Please insert citation pg 2 line 84 is missing the citation (7)

2.     The conclusion steel to be rephrased stating the clear results obtained in this study. 

Reading the abstract it is mandatory to understand the study, the aim, results and conslusion. 

In your conclusion is still missing the main results of the study.

Please consider to write clearly that, according to your study: multifocality, age and capsular invasison are the independent factors that allowed you to establish a scoring system in  PTMC. This system might  be used as an efficient evaluation algorithm in etc….

Author Response

Dear professor,

Thanks for your comments again, and our team has corrected the following problems.

  1. we have inserted citation No.7 in the sentence in pg 2 line 84.
  2. we have rewritten the conclusion.

Best regards,

Cai Wei & Yang Zheyu